# Burden of COVID-19 Pandemic Perceived by Polish Patients with Multiple Sclerosis

**DOI:** 10.3390/jcm10184215

**Published:** 2021-09-17

**Authors:** Anna Pokryszko-Dragan, Justyna Chojdak-Łukasiewicz, Ewa Gruszka, Marcin Pawłowski, Tomasz Pawłowski, Anna Rudkowska-Mytych, Joanna Rymaszewska, Sławomir Budrewicz

**Affiliations:** 1Department of Neurology, Wroclaw Medical University, Borowska 213, 50-556 Wroclaw, Poland; anna.pokryszko-dragan@umed.wroc.pl (A.P.-D.); ewa.gruszka@umed.wroc.pl (E.G.); slawomir.budrewicz@umed.wroc.pl (S.B.); 2Department of Psychiatry, Wroclaw Medical University, Wybrzeże Pasteura 10, 50-367 Wroclaw, Poland; marcin.pawlowski@student.umed.wroc.pl (M.P.); tomasz.pawlowski@umed.wroc.pl (T.P.); psychiatria@umed.wroc.pl (A.R.-M.); joanna.rymaszewska@umed.wroc.pl (J.R.)

**Keywords:** multiple sclerosis, COVID-19 pandemic, perceived stress, coping strategies

## Abstract

People with multiple sclerosis (MS) were expected to be particularly affected by the COVID-19 pandemic. The purpose of the study was to evaluate the burden of pandemic, perceived by Polish MS patients, with regard to major contributing factors. The survey, conducted in August/September 2020, included: Perceived Stress Scale (PSS-10), Coping Orientations to Problems Experienced (Brief–COPE), questions on demographic data, MS characteristics, and health-related and social aspects of pandemic burden. Relationships were searched between PSS-10 and Mini-COPE results and other analyzed items, using U Mann–Whitney test, Kruskal–Wallis ANOVA rank test and Spearman rank correlation. The survey was answered by 287 MS patients (208 female, 79 male, aged 21–69 years). Since March 2020, 2.4% of respondents had been positive for COVID-19 and 5.2% had undergone a quarantine. Mean PSS-10 score was 19.99, with moderate or high level of stress in 83.3% of respondents. Problem-focused strategies were more frequently used than emotion-focused strategies (1.76 vs. 1.16). Higher PSS-10 score was associated with comorbidities (H = 4.28), increase in major MS symptoms during the pandemic (21.92 vs. 18.06), experience of healthcare limitations (21.12 vs. 17.98), work-related (22.58 vs. 18.69), financial (22.70 vs. 18.83) and family-related problems (22.54 vs. 17.73) due to pandemic restrictions. A coping model was associated with functional disability and limitations to daily activities (H = 7.81). During the first stage of the pandemic, MS patients reported increased level of stress and preferred problem-focused coping. The level of stress and coping showed more relationships with pandemic impact upon social issues than with MS-related variables.

## 1. Introduction

The outbreak of novel coronavirus disease (COVID-19), declared by the WHO as a pandemic in March 2020, has had a profound and unprecedented impact upon health and social issues worldwide. Severe restrictions (closing of educational and cultural institutions, limitations to service facilities, recommended social distance) were instituted to mitigate rapid spreading of COVID-19 disease. Healthcare settings have been undergoing massive reorganization, prioritizing management of COVID-19 and providing safety for patients and staff, which resulted in interruption or limitation to certain services. A fear of life-threatening infection, accompanied by a rapid change in lifestyle and socio-economic problems, undoubtedly has had an adverse impact upon well-being of people throughout many countries.

Multiple sclerosis (MS) is a chronic, immune-mediated disease which affects the central nervous system, leading to its long-lasting and disseminated damage. Demyelinative lesions and axonal loss within brain and spinal cord cause multifocal symptoms and signs of neurological deficit. People with MS were expected to be particularly affected by the pandemic. Disability and immunocompromised status due to disease-modifying treatment (DMT) were initially considered as predisposing factors for severe course to various forms of therapy and rehabilitation [1,2]. Furthermore, cognitive and mental health problems which often occur in MS population, together with insufficient coping abilities, would make these patients more vulnerable to distress caused by the pandemic [3,4].

In Poland, the lockdown was introduced in March, with the restrictions gradually canceled during the summer months. The statement of experts from Polish Neurological Society, concerning management of MS during the pandemic, was published in March and updated in May [5]. The document included recommendations with regard to prevention and treatment of COVID-19 infection in MS population, safety issues at health care facilities, as well as initiation and continuation of DMT. National MS Society also aimed at providing information and support to the patients and their caregivers. Nevertheless, the pandemic has undoubtedly affected MS population in many aspects. Recognition of this impact from the patients’ perspective seemed essential for addressing their needs and effective management of major problems emerging from pandemic consequences [6].

The purpose of the study was to evaluate the burden of the pandemic, perceived by Polish MS patients, including the level of experienced stress and coping strategies. We also aimed at identification of the factors most related to stress and coping, including demographics, MS-related and social issues.

## 2. Materials and Methods

The study questionnaire, based on relevant literature, was created by neurologists and psychologists experienced in MS (Appendix A). The following sections were included:Sociodemographic data;Exposure to COVID-19 infection;MS-related items: general clinical characteristics of the disease and its course during the pandemic;Social aspects of the pandemic burden.

The participants were asked to report all the aspects of the pandemic burden experienced from March 2020 to the time of responding to the survey.

Perceived Stress Scale (PSS-10) [7] was introduced to measure a level of experienced stress and the shortened version of Coping Orientations to Problems Experienced (Brief–COPE) [8]—to evaluate coping strategies. Both tools were provided in the standardized Polish version [9].

PSS-10 [7,9] contains 10 items which refer to subjective perception of situations and problems within the preceding 4 weeks. Response to each item can be scored 0–4 and their sum makes the total score, ranging from 0 to 40. Higher score indicates greater level of perceived stress. According to a 10-degree sten scale, a total score within stens 1–4 reflects low level of stress, between 5 and 6 sten—moderate level, and above 7—high level of stress.

Brief-COPE [8,9] consists of 28 items, associated with 14 coping strategies. The response to each item is rated 1–4 and the average score is calculated for each coping strategy, indicating its utilization by the subject. Coping strategies can be also categorized into their main types.

The survey was conducted parallel through online and printed modes in August and September 2020. Computer-assisted web interviewing (CAWI), an Internet surveying technique in which the interviewee follows a script provided in a website, allowed to create Google Forms, with the link posted on the central website of Polish MS Society and throughout social media profiles of its regional divisions.

The printed version of the same questionnaire was freely distributed among the willing subjects visiting the two major outpatient clinics in Lower Silesia (south-western region of Poland). These subjects were informed about two eligible versions of survey and were requested to submit only one of them. Completed printed questionnaires were collected by medical assistants not involved in the research team. After six weeks the number of respondents to the online survey reached 128 and no new responses were recorded within the next two weeks. By then, 159 complete responses to the printed questionnaires had been collected and the database was closed.

The responses collected by CAWI method were downloaded, the responses from printed questionnaires were recorded as the text files and all the data were transferred into Microsoft Excel (Microsoft Corporation, Redmond, WA, USA) spread sheet for statistical analysis.

Mean values with standard deviation were calculated for PSS-10 and Brief-COPE scores, and the distribution of responses in particular categories was analyzed. Relationships were searched between PSS-10 and Brief-COPE results and other analyzed variables.

To select proper statistical methods, a normality of distribution for all continuous variables was verified with a Shapiro–Wilk test. For the majority of variables, an assumption of normal distribution was not met. Therefore, non-parametric tests were used for the analysis. Comparisons between two independent groups were performed with U Mann–Whitney test or t-Student test if the proper assumptions were met. In case of comparisons for which an independent variable was of a categorical type, a Kruskal–Wallis ANOVA rank test was applied. To assess the correlations, the Spearman rank correlation was applied. To assess the independence of categorical variables, Pearson Chi-square test was used. A *p*-value less than 0.05 was considered statistically significant. The statistical analysis was performed using STATISTICA PL v.8 statistical software (StatSoft, Kraków, Poland).

The project of the study was approved by Local Bioethical Committee at Wroclaw Medical University. Participation in the survey was voluntary and without any financial compensation. Anonymity of the responses was maintained throughout collection and storage of data. An informed consent form to participate in the study and allow procession of data for research purposes was provided in the initial part of the questionnaire. Its confirmation in the online version was necessary to proceed with responding to the questions and submit the filled questionnaire. In the printed version, consent was confirmed by signing the form and completion of the questionnaire.

## 3. Results

### 3.1. Characteristics of the Study Group

The study comprised 287 people with MS (208 female, 79 male, aged 21–69 years, mean 41.05). Among this number, 128 people (97 females, 31 males, mean age 36.44 years) responded through the online version of the survey and 159 people (111 females, 48 males, mean age 44.76 years) responded through the printed version. Table 1 presents the demographic characteristics of respondents.

Duration of MS in the study group ranged between 1 and 40 years (mean: 11.8). The majority of patients (165) had relapsing-remitting MS, although 30% could not define the type of disease. Among the major MS-related complaints, the most frequent included fatigue, disturbance of gait, balance problems and vertigo (Table 2). Minor or mild functional disability with unlimited distance of ambulation was declared by 86.4% of respondents. DMT were used in 81.9% of patients and only symptomatic treatment—in 11.8% (Table 2). Comorbidities reported by 121 respondents included: endocrine disorders—thyroid gland diseases, diabetes (38), cardiovascular system diseases—hypertension, coronary artery disease (35), bronchial asthma (12), psychiatric conditions—depression, anxiety disorders (10), spondyloarthrosis (8), other immune-mediated diseases (rheumatoid arthritis, psoriasis) (6), endometriosis (6), and glaucoma (6).

Since the onset of the pandemic (March 2020), 73 patients (25.4%) experienced a relapse, which in 64 cases was confirmed by consulting neurologist. Out of these patients, 50 were treated with i.v. infusions of corticosteroids (in- or outpatient schedule) and 14 with oral medications. Within this timeframe, 144 of respondents (50.2%) reported an increase in frequency and/or severity of previously experienced major MS symptoms, and 27 (9.4%) observed some new symptoms, including headache, mood disturbances and insomnia.

The respondents to the printed version of the survey less often had relapsing-remitting MS and more often could not define MS type in comparison to those who responded online (45.9% RRMS, 42.1% undefined vs. 71.88% RRMS, 16.6% undefined, respectively, *p* < 0.0001). The online respondents more often experienced an increase in their major MS symptoms (60.94% vs. 41.5%, *p* = 0.001). No other differences in MS-related issues were found between the subgroups of participants.

### 3.2. Exposure to COVID-19

Since March 2020, out of 46 (16%) patients who had been tested for COVID-19, 7 (2.4%) were positive and 3 of them needed hospitalization. Fifteen patients (5.2%) had undergone a quarantine, while 19 (6.6%) reported COVID-19 infection in a family member or a close person. No significant difference in occurrence of relapse was found between the subjects exposed to COVID-19 and those who were not (20.06% vs. 19.98%). The subgroups of respondents to the online and printed version of the survey did not differ in reported exposure to COVID-19.

### 3.3. Health Care-Related Impact of the Pandemic

Problems with the access to a neurologist were reported by 63 (21.9%) respondents, with the access to a primary care physician or other specialist in 144 (50.2%) cases, and with access to rehabilitation in 103(35.9%). Thirty four (11.8%) subjects found relevant information about their condition hardly available. With regard to DMT, 29 (10.1%) patients reported the obstacles in continuation of scheduled treatment, 16 (5.6%)—a delay in introduction of treatment and 8 (2.8%)—a delay in planned switch to another DMT. Forty-five persons (15.7%) cancelled in- or outpatient visit because of fear of infection, and 34 of them were offered a remote consultation.

### 3.4. Social-Related Impact of the Pandemic

Work-related problems were reported by 96 patients (33%): 40 had their job suspended, 14 were fired, 14 experienced difficulties in turning to remote work, 8 complained of increased workload and 20 expressed fear of infection at workplace. Eighty-six subjects (30%) experienced financial problems.

Family-related problems were reported by 134 patients (46.7%) and included a concern about the health of family members, limited contacts due to epidemic restrictions, need to help children with online learning, compromising remote work with family issues and other conflict situations associated with staying at home during lockdown.

Due to the pandemic restrictions, 52.6% of respondents had to cancel or re-schedule an important life event. Moreover, 129 (44.9%) respondents reported problems with daily activities (shopping, household duties, small repairs, pet care) and 20 (6.9%) respondents needed extra help/support in this field.

### 3.5. PSS-10 and Brief-COPE Results

The mean PSS-10 score in the study group was 19.99 (range: 1–40). According to the sten scale, 48 subjects (16.7%) had a low level of perceived stress, 80 (27.9%) with a moderate level and 159 (55.4%) with a high level (Figure 1).

The mean PSS-10 score in the respondents to the online version of the survey was significantly higher than in the respondents to the printed version (21.78, SD 6.8 vs. 18.56, SD 6.77, *p* < 0.0001).

According to the Brief-COPE results from the study group, the most commonly used coping strategies were: acceptance, planning and positive reframing. Substance use, behavioral disengagement, religious coping or denial were the least frequently used ones (Figure 2). With regard to the main categories of coping, problem-focused strategies were more preferred than emotion-focused ones (1.76 vs. 1.16). (Figure 2). The respondents to the online version of the survey less often used emotion-focused coping strategies than the respondents to the printed version (1.07 vs. 1.24, *p* = 0.01).

### 3.6. Relationships of PSS-10 and Brief-COPE Results with Other Variables

The PSS-10 score negatively correlated with age (R = −0.150, *p* = 0.011) and was higher in females than in males (20.94 vs. 17.52, *p* < 0.0001). The patients who declared increased frequency/severity of their major MS-related symptoms, had higher PSS-10 score than the remaining ones (21.92 vs. 18.06, *p* < 0.0001), and such significant difference was observed for those with comorbidities (*p* = 0.0386). No other relationships were found between PSS-10 score and demographic or clinical factors.

Significantly higher level of stress in PSS-10 was associated with health care-related shortcomings (21.12 vs. 17.98, *p* = 0.0010), work-related problems (22.58 vs. 18.69, *p* = 0.000024), financial difficulties (22.70 vs. 18.83, *p* = 0.000024), family-related problems (22.54 vs. 17.73, *p* = 0.000000) and a need for extra help in daily activities (23.65 vs. 19.72, *p* = 0.0155).

There was a significant negative correlation between level of stress in PSS-10 and a preference for problem-focused coping strategies (R = −0.1293, *p* = 0.0284).

Emotion-focused strategies were more frequently used by females (1.23 vs. 0.97, *p* = 0.0012). The respondents with moderate and severe functional disability more often than the others used problem-focused strategies (*p* = 0.049). The patients who experienced increase in major MS symptoms, less often had substance use (*p* = 0.0428), behavioral disengagement (*p* = 0.0032) and self-blame strategies (*p* = 0.0395) than the remaining ones. A need for help in daily living was significantly associated with the use of problem-focused (2.04 vs. 1.42, *p* = 0.0401) and emotion-focused strategies (1.42 vs. 1.14, *p* = 0.0378).

No other relationships were found between the PSS-10 or Brief-COPE results and demographic, clinical or pandemic-related factors.

## 4. Discussion

The COVID-19 pandemic was demonstrated to cause an increased level of anxiety and depression, and a worse sleep quality in MS patients throughout different countries [2,10,11,12,13]. The Italian authors of [14] reported a greater prevalence of severe stress (measured in PSS-10) in MS subjects, in comparison with the healthy controls or the patients with migraine. The results of PSS-10 in our study indicate a moderate or high level of perceived stress in a vast majority of respondents (more than 80%). The international cohort surveys, conducted during the pandemic, also revealed a moderate to high level of stress in healthy adults, with mean PSS-10 score up to 19 points [15,16]. MS subjects are considered to have greater susceptibility to stress, due to the background of the disease (autoimmune response associated with autonomic and endocrine dysregulation) and its specificity (long-lasting and unpredictable course, accumulating disability). However, a diverse range of PSS-10 score had been obtained in the studies in this field [17,18,19,20,21,22], which suggests individual differences in perception of stress in MS population. With regard to the pandemic, no comparative studies were conducted with the use of PSS-10, but a prevalence of anxiety and depression in MS patients did not increase in comparison with the pre-pandemic assessment [10,23]. Thus, it remains disputable whether the pandemic has indeed enhanced distress experienced by MS patients.

There is some evidence [3,24,25,26] that people with MS tend to undertake passive and emotion-focused coping strategies, which makes them more vulnerable to stressful life events. However, adaptation to the limitations posed by a disease, as well as psychological support, may cause a shift towards more active and effective coping. The Brief-COPE results showed that the respondents to our survey overall preferred problem-focused coping, with planning, active coping and positive reframing as the most frequently used strategies. In another study, Spanish MS patients, surveyed during the pandemic, favored active confrontation and religion, while their use of emotional support, humor and positive re-evaluation was less frequent than in healthy controls [4]. The Italian survey [13] also indicated preference for positive attitude, problem solving and turning to religion among MS patients. It is worth highlighting that epidemic restrictions significantly limited the access to the sources of instrumental (from health-care and rehabilitation facilities) and emotional (contacts with family and friends) support which provide a basis for relevant coping strategies [13]. MS patients perceived especially a decrease in social support as a major negative impact of lockdown [3,14]. In healthy adults, positive coping strategies were shown to moderate a distress caused by the pandemic [27,28], but it was suggested that preferred strategies might reflect a temporary reaction to an unprecedented traumatic situation, and not necessarily a usual coping model [27]. In our study group, the increased level of stress was associated with a weaker preference for problem-focused coping. Supposedly coping strategies in MS subjects have developed in the long-term course of disease but were additionally affected by a temporary pandemic distress.

### 4.1. Demographic and Social Factors

In the study group, the perceived level of stress was higher in younger and female patients, and the latter used emotion-focused coping strategies more often than males. Similar relationship between age and distress caused by the pandemic was observed in MS subjects [1,11,12,29] and in healthy adults [15,16]. Although elderly persons are regarded to be at greater risk of severe COVID-19 infection, their perception of stress is probably modified by memory of past experiences and regulation of emotion [16,29]. Moreover, in comparison with young adults, they are less burdened with work or family obligations and less frequently use internet social media as a source of information about the pandemic [1,12]. It is worth considering that both higher PSS-10 score and lower mean age were found in those who responded to the online version of our survey, in comparison with respondents to the printed version. Sex differences in perception of pandemic stress were consistently reported for general populations [15,16,27]. Although female and male MS subjects presented with a similar level of depression and anxiety [11], women more often expressed fear of COVID-19 infection and tended to avoid exposure [30]. It should be noted that due to greater disease prevalence in women, they constitute the majority of all the investigated MS groups, which may affect the findings.

Other demographic factors, including residence and education level, did not affect the level of stress or coping model in our MS group. Apart from the lower level of anxiety in MS patients with academic degree [11], no such relationships were found by other authors, either. However, a high proportion of those with completed university education among the respondents to our survey might have had some impact upon the results, e.g., with regard to coping. Although vocational status was not related to stress or coping, those of our respondents who experienced work-related or financial problems (ca. 1/3) due to epidemic restrictions declared significantly higher levels of stress. The pandemic’s impact on the employment situation of our respondents included changes in type or schedule of work, as well as loss of job. Even more detrimental consequences were reported for MS patients in U.S. [30,31]. In the Italian MS subjects, unemployment during the pandemic significantly contributed to depression and anxiety [2]. Changes in employment (including remote work) and lower income level were considered as factors substantially mediating pandemic stress in healthy adults [15]. Vocational activity is an important element of MS patients’ social functioning, often adversely affected by the disease. Current but also long-term work-related problems, resulting from pandemic restrictions, might significantly worsen their economic situation as well as psychosocial condition [31]. These issues should be addressed with adequate system of support and counseling.

Similar observations concerned a family situation of MS subjects in the study group. Level of stress and coping preferences did not depend on marital status or dependents, but the pandemic impact upon family life, reported by almost half of the respondents, caused an increased level of stress. Family-related concerns included emotional issues, as well as logistic problems with consequences of lockdown. A forced social distance which prevented relationships was shown to affect stress and well-being in MS patients [14]. On the other hand, more time spent at home with family or partners resulted in positive impacts on their mood and sexual satisfaction [23].

### 4.2. Exposure to COVID-19 Infection

Among the respondents to our survey, only a small percentage have become infected or exposed to COVID-19. In comparison with other countries, during the first months of the pandemic, the prevalence of COVID-19 and morbidity in Poland were relatively low. Furthermore, MS patients presumably considered themselves at greater risk of infection and thus were undertaking more preventive behavior [14]. The direct exposure to infection in our MS patients was not associated with higher level of stress, similarly to the findings about depression and anxiety, reported by Altschuler et al. [1]. However, those with comorbidities, supposedly increasing the risk of severe COVID-19, presented with higher PSS-10 score. In healthy adults, symptoms suggestive of COVID-19 or perceived increased susceptibility to infection were shown to increase level of stress [16,27]. It may be hypothesized that MS subjects, due to long-lasting disease, are more adapted to concerns about their health and perceived vulnerability [23].

### 4.3. MS Related Factors

Only 25% of the study group experienced a relapse during the pandemic, while more than a half reported an increased frequency or severity of their major MS-related complaints. In the study of Motolese et al. [2], only 20% of the patients experienced new or enhanced MS symptoms (mainly sensory impairment and fatigue), while the other authors did not focus on occurrence of relapses during the pandemic.

Our MS patients who reported an increase in chronic symptoms declared a higher level of perceived stress. However, there was no difference in PSS-10 results between those who had relapsed or not. Stress was often considered as a possible trigger of clinical or radiological MS activity, as well as the effect of adverse outcome of the disease [32,33,34,35,36,37,38,39,40]. A diversity of results from particular studies suggests a complexity of links between stress and MS course, including a “vicious cycle” mechanism. There are also contradictory findings from studies investigating the course of MS during extreme traumatic situations. No increase in relapses was observed in Japanese patients following the great earthquake in 2016 [40] or in Israeli patients exposed to Persian Gulf war in 1991 [41]. On the contrary, more frequent relapses (associated with greater subjective stress) were noted during the Hezbollah–Israel war in 2006, with life threat and displacement identified as the main sources of stress [42]. During the COVID-19 pandemic, a higher anxiety level was shown in the Egyptian patients with relapses [12] while no such association was confirmed for the U.S. ones [1]. It seems that major stressful events do not directly affect MS activity, but their impact is mediated by individual perception, as well as by support provided to the patients [43].

Interestingly, no significant relationships were found in the study group between level of stress or coping model and any MS-related measures. Only more severe disability (assisted walk) was associated with greater preference for problem-focused coping strategies, and the level of stress was higher for those who declared the need for extra help during pandemic. Thus, it seems that the impact upon social functioning was more related to stress and coping than the disease itself. It should be highlighted that the vast majority of our respondents declared minor or mild disability and relapsing-remitting type of MS. Furthermore, in comparison with other studies [1,13,14], relatively small percentage of our patients received second-line DMT (apparently due to more benign or stable course of disease). Perhaps those with more active MS and greater disability, who did not participate in the survey, would perceive themselves as more endangered by severe COVID-19 infection, which could have affected the overall results.

With regard to the management of health issues, ca. 20% of our respondents had problems with the access to a neurologist and initiating or continuing DMT, while almost a half complained of limited availability of rehabilitation or primary health care facilities. Otherwise, there was a small proportion of those who canceled their visits due to fear of infection. The studies conducted in U.S. MS populations demonstrated greater disturbances in health care services, both from the providers’ and patients’ side [30,31] Throughout the countries, appropriate preventive measures were arranged for MS care centers [44,45]. For reimbursement reasons, DMT in Poland are provided under the charge of the specialist centers, within the unified schedule supervised by National Health Fund. Thus, the main framework for the treatment and its monitoring was usually maintained, although local disruptions might have occurred due to health-related or organizational consequences of the pandemic. According to national recommendations for health care settings and the statement of Polish MS experts [5], safety measures were undertaken at MS centers: e.g., the schedule of visits adjusted to maintain distance, obligatory use of masks and hand disinfection, screening for symptoms of infection at the entrance. Depending on local resources, remote consultations were being arranged (phone calls, video calls, sending comments and diagnostic tests results via e-mail). Our respondents seemed satisfied with a specialist care and with information on their current situation, offered probably also by the patients’ organizations.

### 4.4. Limitations and Strengths

Several limitations to this study should be considered, including a lack of a comparator group of healthy controls or pre-pandemic assessment of stress and coping in MS subjects. The study group cannot be treated as population-based because of its moderate size and some bias it was subjected to. The online survey (though distributed nationwide) favored those who have access to internet and are interested in activity of MS patients’ organizations, while the printed version was distributed only in the two major outpatient centers in one region. A comparison of findings from these two subgroups of respondents revealed differences mainly in demographic but not MS-related items. Therefore, considering the moderate number of participants, we did not conduct further analyses for each subgroup separately.

Furthermore, the diagnosis of MS and disease-related issues were not verified through medical records or physician’s opinion but only based on self-observation, which might have posed some ambiguity. The items on health-related issues either included a range of options to choose from (major MS symptoms, type of disease, treatment regimen) or allowed free-text answers (coexisting diseases, new symptoms), which might have also affected the results.

However, we believe that the findings from this study allow a better insight into MS patients’ perspective of pandemic burden. Lessons learned from their experience may contribute to identification of the major aspects of pandemic impact, including some concealed and indirectly harmful issues, and expected long-term consequences. These aspects need to be addressed during the ongoing pandemic, but also in future plans of effective MS management [6,10,32]. Despite relatively preserved medical services, the patients suffered mainly from a disruption in holistic system of their support and pandemic effects on their economic status and daily living. Thus, informational and educational materials should focus not only on the current pandemic or MS-related issues, but also concern healthy lifestyle, effective coping and maintaining relationships [6,11]. Pandemic experiences might encourage a wider use of innovations: e.g., telehealth consultations, online physiotherapy programs. In view of long-term pandemic consequences, support and counseling should be provided in the field of mental well-being, as well as economic and work-related aspects [6,31].

## 5. Conclusions

Polish MS patients surveyed during the first stage of the COVID-19 pandemic presented with a moderate to high level of perceived stress and preferred problem-focused coping strategies. The burden of the pandemic perceived by the patients was associated mainly with their social functioning and, to a lesser degree, with health status or health care services. The level of stress and coping profile showed more relationships with pandemic impact upon social issues than with disease-related variables. Consequences of the pandemic should be addressed with adequate support and counseling in management of MS patients.

## Figures and Tables

**Figure 1 jcm-10-04215-f001:**
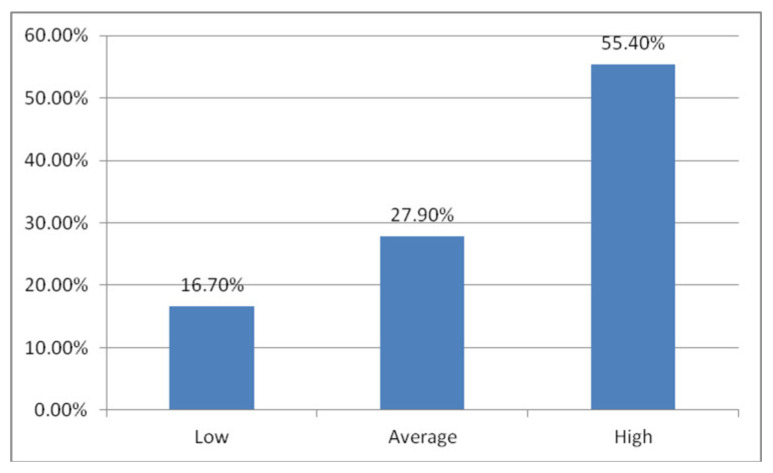
Distribution of PSS-10 results (sten scale indicating level of perceived stress) in the study group.

**Figure 2 jcm-10-04215-f002:**
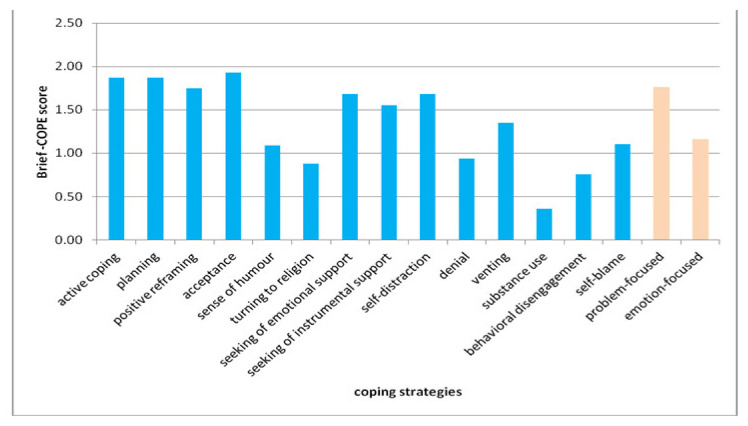
Use of coping strategies according to the Brief-COPE results in the study group.

**Table 1 jcm-10-04215-t001:** Demographic characteristics of the study group.

	All Group *n* = 287	Paper Version *n* = 159	On-Line Version *n* = 128
Marital status			
married	155 (54%)	97	58
single	54 (18.8%)	27	27
living together	56 (19.5%)	20	36
divorced	14 (4.9%)	9	5
widowed	6 (2.1%)	5	1
separated	2 (0.7%)	1	1
Children			
No	126 (43.9%)	51	75
Yes	161 (56.1%)	108	53
number of children: 1	78	52	26
2	71	49	22
3	9	6	3
4	3	1	2
Living environment			
city	148 (51.6%)	73	75
small town	80 (27.9%)	45	35
village	59 (20.5%)	41	18
Education level			
primary school	5 (1.7%)	4	1
vocational school	16 (5.6%)	13	3
high school diploma	110 (38.3%)	70	40
university degree	156 (54.4%)	72	84
Occupation status			
student	9 (3.1%)	0	9
unemployed	18 (6.3%)	12	6
employed (full-time job)	147 (51.2%)	71	76
employed (part-time job)	21 (7.3%)	10	11
self-employed	14 (4.9%)	11	3
retired	17 (5.9%)	14	3
disability pensioner	61 (21.3%)	41	20

**Table 2 jcm-10-04215-t002:** Clinical characteristics of the study group.

Type of MS	
relapsing-remitting (RRMS)	165 (57.5%)
primary or secondary progressive (PPMS/SPMS)	35 (12.2%)
undefined	87 (30.3%)
Functional disability level	
minor (unlimited mobility)	130 (45.3%)
mild (limited ambulation distance)	118 (41.1%)
moderate (need for assistive devices)	33 (11.5%)
severe (using wheelchair)	6 (2.1%)
Most debilitating MS symptoms	
fatigue	123 (42.9%)
gait disturbance	112 (39%)
vertigo/disturbed balance	83 (28.9%)
upper limbs dysfunction	68 (23.7%)
sensory impairment	68 (23.7%)
bladder dysfunction	57 (19.9%)
visual impairment	57 (19.9%)
cognitive decline	54 (18.8%)
Disease modifying therapy (DMT) *n* = 235	
dimethylfumarate	74 (31.5%)
IFN ß-1a	44 (18.7%)
IFN ß-1b	37 (15.7%)
fingolimod	29 (12.3%)
teriflunomide	22(9.4%)
glatiramer acetate	13 (5.5%)
natalizumab	9 (3.8%)
ocrelizumab	2 (0.9%)
cladribine	1(0.4%)
experimental therapy in clinical trials	3 (1.3)
immunosuppressive treatment (azatioprine)	1 (0.4%)

## Data Availability

The data presented in this study are available on request from the corresponding author.

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
