# Peer review of "Burden of COVID-19 Pandemic Perceived by Polish Patients with Multiple Sclerosis"

_jcm, 2021, doi:10.3390/jcm10184215_

Round 1

Reviewer 1 Report

Authors have addressed my concerns.

Reviewer 2 Report

Thanks so much to the authors. I am happy for this to proceed. 

This manuscript is a resubmission of an earlier submission. The following is a list of the peer review reports and author responses from that submission.

Round 1

Reviewer 1 Report

Pokryszko-Dragan and colleagues reported on coping, functional disability and limitations to daily activities in Polish patients with multiple sclerosis (MS). The topic is interesting, though largely covered by medical literature. The manuscript is overall clear and well written. However, I have some major comments to the authors, and I would also recommend the authors better compared their findings to other international experiences.

It is not mentioned how MS patients were recruited. Consecutive recruitment for patients attending one MS clinics? If so, this could represent a selection bias, with the inclusion of patients who actually attended the MS centre, while authors excluded those who remained at home and, thus, more likely affected. I believe recruitment and potential biases should be carefully described and discussed.

Also, how was sample size determined?

There is a very small proportion of individuals using second line treatments. Can authors comment on this? How much would this population relate to other studies?

For figure 2, the legend on the x-axis is missing.

I wonder whether the MS centre put in place measures to mitigate the effects of COVID19 (e.g., Buonomo et al. Mult Scler Relat Disord 2020, and Moccia et al. Neurol Sci 2020). If this is the case, these should be described. Otherwise, possible measures to be implemented should be described and compared to previous experiences.

In the discussion, I believe authors could compare their findings also to Costabile et al. Eur J Neurol 2020.

There are minor English mistakes throughout the text. I.e., persons should read people. Careful revision is recommended.

Author Response

We appreciate the overall positive opinion on our study, as well as the  thorough review aimed at improvement of the manuscript quality. We did our best to consider all the Reviewers' suggestions in the revised version of the manuscript. Particular remarks are addressed below.

Pokryszko-Dragan and colleagues reported on coping, functional disability and limitations to daily activities in Polish patients with multiple sclerosis (MS). The topic is interesting, though largely covered by medical literature. The manuscript is overall clear and well written. However, I have some major comments to the authors, and I would also recommend the authors better compared their findings to other international experiences.

It is not mentioned how MS patients were recruited. Consecutive recruitment for patients attending one MS clinics? If so, this could represent a selection bias, with the inclusion of patients who actually attended the MS centre, while authors excluded those who remained at home and, thus, more likely affected. I believe recruitment and potential biases should be carefully described and discussed.

Also, how was sample size determined?

The process of the survey distribution and recruitment of the participants has been more precisely described and potential biases were discussed. Online survey (eligible also for those who did not attend any MS centre at this time due to disability or fear from infection) was addressed to people with MS throughout Poland, while printed version was distributed only in two centers within one region. Comparison of findings from responses to online or printed version of survey has been provided in the Results and discussed, as well.

There is a very small proportion of individuals using second line treatments. Can authors comment on this? How much would this population relate to other studies?

The use of DMT in the study group was compared to other studies, and potential bias was discussed.

For figure 2, the legend on the x-axis is missing.

XY-axis for the figure 2 was implemented.

I wonder whether the MS centre put in place measures to mitigate the effects of COVID19 (e.g., Buonomo et al. Mult Scler Relat Disord 2020, and Moccia et al. Neurol Sci 2020). If this is the case, these should be described. Otherwise, possible measures to be implemented should be described and compared to previous experiences.

Preventive measures provided at Polish MS centers were described.

In the discussion, I believe authors could compare their findings also to Costabile et al. Eur J Neurol 2020. The study by Costabile et al was cited and commented in Discussion.

There are minor English mistakes throughout the text. I.e., persons should read people. Careful revision is recommended.

Revision was made and errors were corrected.

Reviewer 2 Report

Here, the authors undertook a mixed-methods (online plus print) survey of people with MS in Poland, seeking to evaluate the psychosocial impacts of the first phase of the COVID-19 pandemic in this population and the potential of coping strategies to reduce the negative effects thereof. Amongst 287 people with MS, the authors unsurprisingly found that stress as measured by PSS-10 was positively associated with MS symptoms and comorbidities, and was higher among those with healthcare limitations and work/family/financial problems. Coping strategies tended to problem-focused rather than emotion-focused, and these were associated with lower PSS-10 scores.

This is an interesting study on a worthwhile topic, and a timely one as well. The study, while small in sample, is quite comprehensive in its measures. I have some comments which I hope the authors might consider.

The Abstract leaves one uncertain how the measures applied relate to the COVID pandemic as the results shown merely enumerate participant characteristics and how these relate to PSS-10 score. It only becomes clear in the body text that indeed the results are very much related to the pandemic, questions about clinical severity, healthcare limitations, and problems asked about these as compared to the period before the pandemic. This should be made clear in the Abstract. Also, it is mentioned in the body text about the proportions of people exposed/infected with COVID-19. Given the proposed aim of the study, this information is quite germane and should be included in the Abstract.

Regarding MS symptoms, comorbidities, and medications, please specify whether these are selected by participants from a list or whether this was reported as free-text? If the latter, this is potentially an issue as this would lead to reporting bias and this then should be acknowledged in the Discussion.

Other comments:

Abstract

  • The instrument is the Brief-COPE, not the Mini-COPE.
  • Please specify the statistical methods used to compare measures, e.g., ANOVA and Kruskal-Wallis.
  • Please use the terminology sex and male/female rather than gender and men/women.
  • Please specify what the PSS-10 total score is. I might suggest presenting the mean PSS-10 as 19.99/total.
  • Please specify that the Brief-COPE was used to query coping strategies as this is not clear.
  • Regarding the figures presented for coping strategies, I am unclear if the numbers present (1.76 vs 1.16) are mean numbers of strategies utilised or what these are.
  • Regarding increased major MS symptoms, please specify that this is current vs the time preceding the pandemic.
  • Please specify how comorbidities were queried.
  • Regarding presentation of statistics comparing between groups, I would suggest presenting x vs y with the p-value from the respective statistical method. H statistics are not helpful and I would instead just present all comparisons as I have suggested.
  • Also regarding presentation of statistics between groups, I presume the figures for things like experiencing healthcare limitations or types of problems are percentages? If so, please present as such.

Introduction

  • Please provide a brief (1 sentence is fine) description of what MS is.

Methods

  • Please state what the timeframe in which participants were asked to describe their symptoms. Presumably it was from March 2020 to the time of the study but please specify this.
  • Minor typo – should be Kruskal-Wallis, not Wallies.

Results

  • Please specify that it was the start of the pandemic in Poland in March 2020.
  • Regarding the participants in quarantine, I presume this is at the time of the survey?
  • Regarding relapse rates vs COVID exposure, I presume the figures in brackets are %’s?
  • Word choice fix – suggest you mean canceled in/outpatient visits rather than resigned.
  • For section 3.5, I don’t believe the three-level PSS-10 categorisation is a sten scale.
  • Also in this section, suggest changing the word, declared, to had.
  • Regarding p-values in section 3.6, p-values below 0.001 can simply be presented as p<0.001.
  • Please provide some comment about comparisons between those who completed the online vs print surveys. Possibly could be a supplementary table but text is also fine.

Tables/figures:

  • For Tables 1 and 2, please provide %’s in addition to the number.
  • For Table 1, presume Academic is University?
  • Further to this, the very high proportion of participants who had completed university may have some impact on the generalisability of these results. Some comment about this in the Discussion would be worthwhile.
  • Suggest in Table 1 it should be Occupation status.
  • In Table 2, was the most debilitating MS symptom queried as this or how was it asked? Need to be specific as to terminology. Also, could participants select only one or more than one debilitating symptom?
  • Figure 2 is lacking in any labeling. Please fix this. Also, though this may become apparent after labeling I’m unclear as to the meaning of the different colours.

Author Response

We appreciate the overall positive opinion on our study, as well as the  thorough review aimed at improvement of the manuscript quality. We did our best to consider all the Reviewers' suggestions in the revised version of the manuscript. Particular remarks are addressed below.

 Review 1

Here, the authors undertook a mixed-methods (online plus print) survey of people with MS in Poland, seeking to evaluate the psychosocial impacts of the first phase of the COVID-19 pandemic in this population and the potential of coping strategies to reduce the negative effects thereof. Amongst 287 people with MS, the authors unsurprisingly found that stress as measured by PSS-10 was positively associated with MS symptoms and comorbidities, and was higher among those with healthcare limitations and work/family/financial problems. Coping strategies tended to problem-focused rather than emotion-focused, and these were associated with lower PSS-10 scores.

 This is an interesting study on a worthwhile topic, and a timely one as well. The study, while small in sample, is quite comprehensive in its measures. I have some comments which I hope the authors might consider.

 The Abstract leaves one uncertain how the measures applied relate to the COVID pandemic as the results shown merely enumerate participant characteristics and how these relate to PSS-10 score. It only becomes clear in the body text that indeed the results are very much related to the pandemic, questions about clinical severity, healthcare limitations, and problems asked about these as compared to the period before the pandemic. This should be made clear in the Abstract. Also, it is mentioned in the body text about the proportions of people exposed/infected with COVID-19. Given the proposed aim of the study, this information is quite germane and should be included in the Abstract.

Abstract was completed according to the above remarks.

Regarding MS symptoms, comorbidities, and medications, please specify whether these are selected by participants from a list or whether this was reported as free-text? If the latter, this is potentially an issue as this would lead to reporting bias and this then should be acknowledged in the Discussion.

The questionnaire used in the study was provided in the Appendix and possible reporting bias were addressed as limitations of the study.

 Other comments:

 Abstract

  • The instrument is the Brief-COPE, not the Mini-COPE.
  • Please specify the statistical methods used to compare measures, e.g., ANOVA and Kruskal-Wallis.
  • Please use the terminology sex and male/female rather than gender and men/women.

Abstract was completed and corrected.

  • Please specify what the PSS-10 total score is. I might suggest presenting the mean PSS-10 as 19.99/total.
  • Please specify that the Brief-COPE was used to query coping strategies as this is not clear.

More precise description of PSS-10 and Brief-COPE results rating and interpretation was provided.

  • Regarding the figures presented for coping strategies, I am unclear if the numbers present (1.76 vs 1.16) are mean numbers of strategies utilised or what these are.

The figures presented the scores

  • Regarding increased major MS symptoms, please specify that this is current vs the time preceding the pandemic.
  • Please specify how comorbidities were queried.

As mentioned above, the questionnaire used in the study was provided in the Appendix and possible reporting bias were addressed as limitations of the study.

  • Regarding presentation of statistics comparing between groups, I would suggest presenting x vs y with the p-value from the respective statistical method. H statistics are not helpful and I would instead just present all comparisons as I have suggested.

Corrected

  • Also regarding presentation of statistics between groups, I presume the figures for things like experiencing healthcare limitations or types of problems are percentages? If so, please present as such.

 Introduction

  • Please provide a brief (1 sentence is fine) description of what MS is.

Brief definition of MS was provided

Methods

  • Please state what the timeframe in which participants were asked to describe their symptoms. Presumably it was from March 2020 to the time of the study but please specify this.

The information about the timeframe was completed.

  • Minor typo – should be Kruskal-Wallis, not Wallies. Corrected

Results

  • Please specify that it was the start of the pandemic in Poland in March 2020.

This was specified in the Introduction, Methods and Results.

  • Regarding the participants in quarantine, I presume this is at the time of the survey? Exposure to COVID was reported from March 2020 to the time of the survey.
  • Regarding relapse rates vs COVID exposure, I presume the figures in brackets are %’s? Corrected
  • Word choice fix – suggest you mean canceled in/outpatient visits rather than resigned. Corrected.
  • For section 3.5, I don’t believe the three-level PSS-10 categorisation is a sten scale.

More precise description of PSS-10 and Brief-COPE results rating and interpretation was provided.

  • Also in this section, suggest changing the word, declared, to had. Corrected
  • Regarding p-values in section 3.6, p-values below 0.001 can simply be presented as p<0.001. Corrected
  • Please provide some comment about comparisons between those who completed the online vs print surveys. Possibly could be a supplementary table but text is also fine.

Comparison of the two subgroups of respondents was added to the results (Tab.1 and the text) and the findings were commented in Discussion.

Tables/figures:

  • For Tables 1 and 2, please provide %’s in addition to the number. Corrected
  • For Table 1, presume Academic is University? Corrected
  • Further to this, the very high proportion of participants who had completed university may have some impact on the generalisability of these results. Some comment about this in the Discussion would be worthwhile. This was commented in the Discussion.
  • Suggest in Table 1 it should be Occupation status. Corrected
  • In Table 2, was the most debilitating MS symptom queried as this or how was it asked? Need to be specific as to terminology. Also, could participants select only one or more than one debilitating symptom?

  The questionnaire used in the survey was added as the Appendix

  • Figure 2 is lacking in any labeling. Please fix this. Also, though this may become apparent after labeling I’m unclear as to the meaning of the different colours.

The labeling the figure 2 was implemented.